# Collapsing Glomerulonephritis in a Kidney Transplant Recipient after mRNA SARS-CoV-2 Vaccination

**DOI:** 10.3390/jcm11133651

**Published:** 2022-06-24

**Authors:** Francisco Gómez Preciado, Rosa Alejandra De Carvalho Ovalles, Sergi Codina, Laura Donadeu, Alexandre Favà, Laura Martinez Valenzuela, Diego Sandoval, Bárbara Fernández-Cidón, Oriol Bestard, Pedro Alia-Ramos, Montserrat Gomà, Edoardo Melilli, Josep M Cruzado

**Affiliations:** 1Department of Nephrology, Hospital Universitari de Bellvitge (HUB), 08907 Barcelona, Spain; fgomezp@bellvitgehospital.cat (F.G.P.); scodina@bellvitgehospital.cat (S.C.); afava@bellvitgehospital.cat (A.F.); lmartinezv@bellvitgehospital.cat (L.M.V.); diego.sandoval@bellvitgehospital.cat (D.S.); 2Pathology Department, Hospital Universitari de Bellvitge (HUB), 08907 Barcelona, Spain; rdecarvalho@bellvitgehospital.cat (R.A.D.C.O.); mgoma@bellvitgehospital.cat (M.G.); 3Laboratory of Nephrology and Transplantation, Vall d’Hebron Institut de Recerca (VHIR), Vall d’Hebron Barcelona Hospital Campus, Universitat Autònoma de Barcelona, 08035 Barcelona, Spain; laura.donadeu@vhir.org; 4Clinical Laboratory, IDIBELL-Hospital Universitari de Bellvitge, L’Hospitalet de Llobregat, 08908 Barcelona, Spain; barbara.fernandez@bellvitgehospital.cat (B.F.-C.); palia@bellvitgehospital.cat (P.A.-R.); 5Kidney Transplant Unit, Nephrology Department, Vall d’Hebron Hospital Universitari, 08035 Barcelona, Spain; obestard@vhebron.net; 6Biomedical Research Institute (IDIBELL), Hospital Duran i Reynals, L’Hospitalet de Llobregat, 08907 Barcelona, Spain; 7School of Medicine and Health Sciencies, University of Barcelona, L’Hospitalet de Llobregat, 08907, Barcelona, Spain

**Keywords:** SARS-CoV-2, vaccine, COVID-19, collapsing, glomerulonephritis, mRNA

## Abstract

With the vaccination against severe acute respiratory syndrome coronavirus 2 (SARS-CoV-2), studies are describing cases of glomerulonephritis arising after vaccination. We present the first case of a kidney transplant patient who, after mRNA vaccination against SARS-CoV-2, developed nephrotic proteinuria and renal dysfunction, with a biopsy diagnostic of collapsing glomerulonephritis. No other triggers for this glomerulonephritis were identified. Antibodies against the spike protein were negative, but the patient developed a specific T-cell response. The close time between vaccination and the proteinuria suggests a possible determinant role of vaccination. We should be aware of nephropathies appearing after COVID-19 vaccination in kidney transplant recipients also.

## 1. Introduction

Severe acute respiratory syndrome coronavirus 2 (SARS-CoV-2) infection causes kidney dysfunction through various mechanisms. Both de novo glomerulonephritis and relapses have been reported. Among them, focal segmental glomerulosclerosis (FSGS) has been highlighted, specifically the collapsing glomerulonephritis (CG) variant [1,2,3]. Likewise, administration of the vaccine to the population has revealed cases of glomerulonephritis. There are four published cases of CG after administration of the vaccine in native kidneys, both after mRNA [4] and adenovirus-vector (AdV) vaccination [5]. Recently a case of CG after AdV vaccination in a kidney transplant recipient was also reported, but neither the humoral nor cellular response was studied [6]. We present the first case of a CG in a Caucasian kidney transplant patient developed after the administration of a mRNA COVID-19 vaccine.

## 2. Case Report

He is a 54-year-old man with a history of chronic kidney disease secondary to autosomal dominant polycystic kidney disease, with no other comorbidities. He received a pre-emptive kidney transplant from an unrelated living donor (wife) in 2013. The calculated panel reactive antibody (cPRA) was 0%; Luminex class I and II were negative. Induction was performed with Basiliximab and maintenance with tacrolimus (Advagraf) 4 mg in the morning, mophetil mycophenolate (MMF) 500 mg twice daily and prednisone 5 mg. A 3-month protocol biopsy showed hypertrophic glomeruli with moderate isolated glomerulitis (g2, ai0, at0, ti0, ptc0, av0, aah0, cg0, ci0, ct0, cv0, cm0, C4d0). Immunofluorescence and donor specific antibodies (DSA) were negative. In the last control before the event, he had an estimated glomerular filtration rate (eGFR) of 69 mL/min with a mild proteinuria of 48.6 g/mol, stable since 2017, and stable tacrolimus levels at around 5 µg/L.

The patient received the vaccine for SARS-CoV-2, mRNA-1273, with the first dose on 25 March 2021 and the second dose on 22 April 2021. Ten days after the first dose, he reported the appearance of lower limb oedema, that increased significantly after the second dose. The next day, a urinary dipstick showed high grade proteinuria and the patient promptly contacted our renal transplant unit. On 11 May 2021 we confirmed a nephrotic range proteinuria of 7810 mg/d, with mild creatinine elevation of 120 umol/L, eGFR of 59 mL/min, with plasma albumin in the low limit of 37 g/L and low total proteins of 58 g/L. He had reduced serum IgG (6773 mg/L). He showed indetectable BK, CMV and EBV PCR in blood. DSA were negative. The urine sediment did not show microhematuria. Due to severe nephrotic syndrome, with a proteinuria rising to 13,450 mg/d, a kidney biopsy was performed (Figure 1).

A total of 37 glomeruli were identified, two of which were globally sclerotic. Many of them showed segmental endocapillary hypercellularity occluding lumens and focal foamy macrophage infiltration. In those glomeruli, epithelial hypertrophy and hyperplasia in the Bowman space was seen. Intact portions of the glomeruli showed minimal mesangial expansion, and in other glomeruli collapse of the basement membrane was prominent. Focal acute tubular necrosis was the most relevant finding in the interstitial compartment.

Direct immunofluorescence evaluation was negative in the glomeruli. Additionally, C4d was negative in the peritubular capillaries.

Ultrastructural examination revealed extensive foot process effacement and segmental hypertrophied podocytes. No immune-type electron dense deposits were identified.

With those findings a diagnosis of CG was established. The evolution of kidney function and proteinuria is shown in the Figure 2.

The patient was receiving 4 mg of tacrolimus and had stable levels ranging from 4.7 to 7.6 ug/L. Due to the persistence of proteinuria and oedema, prednisone at 1 mg/kg was started. Afterwards tacrolimus levels decreased slightly to 3.7–5.6 ug/L, but the tacrolimus dosing was unchanged. After evolution he showed partial improvement, with a stabilization of eGFR at 50 mL/min and a marked reduction of proteinuria to 1240 mg/d.

APOL1 gene polymorphism risk variants for FSGS analysis provided negative results (G0/G0). We also performed an evaluation of immune responses at different compartments (serological, memory B cells (mBC) and cytokine (IFN-γ, IL-2, IFN-γ/IL-2) producing T cells) specific to the spike glycoprotein by ELISA and FluoroSpot-based assays, respectively [7]. Detailed information is provided as Appendix A.

Vaccination did not elicit a humoral response in this individual; serologies for SARS-CoV-2 were negative for both anti-N and anti-S IgG (0.67 U/mL), reference value >80). Furthermore, spike specific IgG-producing mBC were not detected.

Nevertheless, T cell reactivity against the spike protein was observed by means of detectable IFN-γ and IL-2 producing specific T cells (11 and seven spots, respectively).

## 3. Discussion

FSGS represents a histological damage that encompasses a variety of conditions [8]. Due to the temporal association with vaccination, our case would be a secondary one.

Secondary FSGS usually presents with proteinuria without nephrotic syndrome, except in specific cases, such as the collapsing variant [9], which stands out due to its etiology and prognosis [10]. Its most common cause is HIV (HIVAN), for which according to animal experimentation [11], viral replication in the kidney seems essential. Furthermore, it is more frequent in African American ethnicity because there is an association with polymorphisms in the gene encoding APOL1 [12].

Other causes of CG are drugs, such as interferon, which induces APOL1 overexpression [13]. Other viruses have also been described as causative agents of CG, such as CMV, Parvovirus B19 and EBV, and their etiopathogenesis have also been related to dysregulation of inflammatory factors, although without clear evidence [14].

During the last year, CG has been defined as the most frequent histopathological diagnosis for COVID-19 patients with acute renal failure or proteinuria [1], giving birth to the term COVID-19 associated nephropathy (COVAN). It has also been described in kidney grafts [2,3,15]. One of the possible mechanisms for this kidney damage is the direct toxicity of the virus since autopsy studies have shown its presence in the kidneys [16]. However, in the cases of COVAN diagnosed by kidney biopsy, the viral presence has not been demonstrated so far, so alternatively it has been suggested that the increase of some cytokines like interferon may be the ultimate cause [17].

Moreover, collapsing glomerulopathy has also been described as a side effect of SARS-CoV-2 vaccination. Cases reported are summarized (Table 1 and Table 2).

The vaccine administered to our patient was mRNA-1273 (Moderna^®^). It is a messenger-RNA vaccine that is delivered in a lipid nanoparticle to be translated into viral protein S.

After the vaccine inoculation, the existence of both humoral, with consequent creation of antibodies, and cellular activation is known [7]. During this process, cytokines such as interferon are secreted. Thus, our hypothesis is that the administration of the vaccine could produce a CG through the secretion of cytokines in patients with unknown predisposing factors. Our patient did not have an allele risk variant for APOL1 protein. Collapsing glomerulopathies have been related to APOL1 risk allele variants, although cases without risk variants have also been described, including in COVAN [18]. Of the patients with collapsing glomerulopathy after SARS-CoV-2 vaccination, one of them was heterozygous for risk variant (G1/G0), which is not associated with increased risk of renal disease.

In our case, the patient did not present antibody titers after vaccination, a frequent finding in transplant patients. In this case the immunoglobulin decrease of nephrotic syndrome might be the reason. However, we identified memory T cells, showing an immune response to the vaccine.

We have not identified other triggering causes of CG in this patient. Although it could be an idiopathic case of CG, due to the temporal sequence, we believe it is secondary to the SARS-CoV-2 vaccine. In any case, the unicity of our case is reassuring us about the safety of vaccination against SARS-CoV-2 in the kidney transplant population.

## Figures and Tables

**Figure 1 jcm-11-03651-f001:**
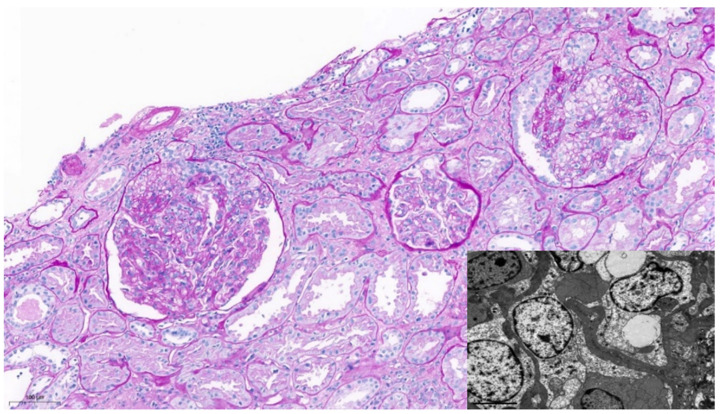
Periodic acid–Schiff stain x13: Endocapillary hypercellularity with epithelial hypertrophy in two glomeruli and segmental foam cells infiltration (**top right**). Basement membrane collapse. No necrotizing lesions were present. The interstitium shows mild focal infiltrates predominantly consisting of lymphocytes and plasma cells. No tubulitis, capillaritis or endothelialitis are present. The degree of interstitial fibrosis and tubular atrophy is mild. Electron microscopy: Extensive foot process effacement and segmental hypertrophied podocytes (**lower left**). Although some of the findings are consistent with the cellular variant, this image is, according to the Columbia classification, a Collapsing Glomerulopathy.

**Figure 2 jcm-11-03651-f002:**
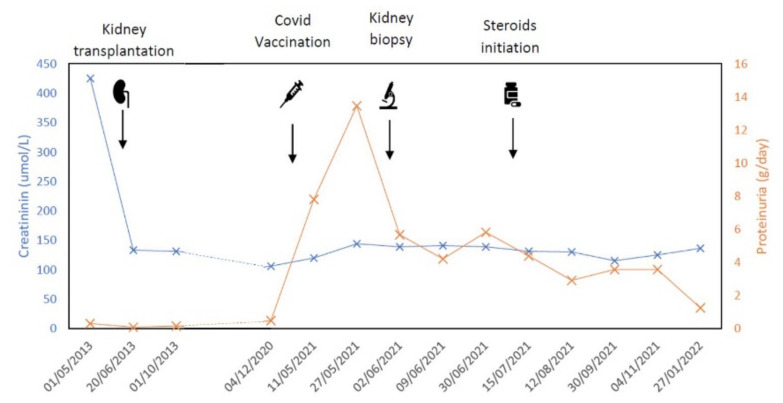
Creatinine and proteinuria evolution through time and defining events. The X axis shows the dates of analysis. The Y axis (left) shows creatinine, and the Y axis (right) shows proteinuria.

**Table 1 jcm-11-03651-t001:** Cases of Collapsing Glomerulonephritis following SARS-CoV-2 vaccination. Baseline characteristics.

Case	Ref.	Age	Sex	Ethnicity	APOL1 Risk Alleles	Kidney Transplant Recipient	Comorbidities
1	Caza et al. [4]	67	F	Black	G1/G1	No	N/S
2	Caza et al. [4]	26	F	Black	G1/G2	No	N/S
3	Neves et al. [5]	63	F	Black	G1/G0	No	Hypertension, heart failure, dyslipidemia
4	Neves et al. [5]	58	F	N/S	G2/G2	No	Multiple myeloma with cast nephropathy
5	Jefferis et al. [6]	N/S	F	N/S	N/S	Yes	N/S

F: Female, N/S: Not specified.

**Table 2 jcm-11-03651-t002:** Cases of Collapsing Glomerulonephritis following SARS-CoV-2 vaccination. Presentation.

Case	Vaccine Type	Dose Number	Renal Presentation	Baseline Creatinine (mg/dL)	Presentation Creatinine (mg/dL)	Renal Outcome
1	Moderna (RNA)	2	NS, AKI	Normal	6.7	Dialysis
2	Moderna (RNA)	2	NS, AKI	Normal	7.7	Partial remission
3	ChAdOx1 (AdV)	1	NS	Normal	0.88	Complete remission
4	ChAdOx1 (AdV)	1	NS, AKI	2–3	N/S	Dialysis
5	ChAdOx1 (AdV)	1	NS, AKI	1.41–2.11	6.03	Dialysis

ChAdOx1: Oxford/AstraZeneca, AdV: Adenovirus based, NS: Nephrotic syndrome, AKI: Acute kidney injury, N/S: Not specified.

## Data Availability

The data that support the findings of this study are available from the corresponding author upon reasonable request.

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
