# Peer review of "Collapsing Glomerulonephritis in a Kidney Transplant Recipient after mRNA SARS-CoV-2 Vaccination"

_jcm, 2022, doi:10.3390/jcm11133651_

Round 1
Reviewer 1 Report
The description of the presented case is quite interesting, however, some issues, especially in the field of pharmacotherapy, are described in a rudimentary way and a vague manner. There is no information on the dosage of immunosuppressants, whether they were modified at the rate of renal deterioration, or adjusted to concentration measurements. Some linguistic errors (typos, stylistic errors) in the manuscript require correction. To sum up, the work is interesting and should be consider with a little revision.
Author Response
Dear reviewer,
Thank you very much for your revision.
The dosage of MMF and tacrolimus has been added, also the range of levels of tacrolimus in blood has been added.
"Induction was performed with Basiliximab and maintenance with tacrolimus (Advagraf) 4 mg in the morning, mophetil mycophenolate (MMF) 500 mg twice daily and prednisone 5 mg."
"The patient was receiving 4 mg of tacrolimus and had stable levels ranging from 4.7 to 7.6 ug/L. Due to the persistence of proteinuria and oedema, prednisone at 1 mg/kg was started. Afterwards tacrolimus levels decreased slightly to 3.7-5.6 ug/L, but the tacrolimus dose was unchanged."
English and other typos have also been corrected.
Reviewer 2 Report
This case report outlines 1 patient with collapsing GN (CG) following COVID vaccination.
1. The literature review seems incomplete. This is not the first case of CG in a kidney transplant recipient, although this case was active COVID infection: https://doi.org/10.1681/ASN.2020060804.
This case report would benefit from a table to outline the literature on CG and COVID infection and vaccination.
2. Did this patient have other comorbidties? Was he tested for active COVID infection?
3. The term COVAN seems to be reserved for those wit 2 APOL1 risk variants. You mention that your patient's analysis was negative, so zero risk variants were determined? This needs clarification and the discussion needs to relate this patient to other cases of CG and how they compare and contrast.
Author Response
Dear reviewer,
Thank you very much for your review. I will address all the comments.
- Indeed, there are other cases of Collapsing Glomerulopathy in Sars-Cov2 infection, also in kidney transplant recipients, as you point in https://doi.org/10.1681/ASN.2020060804 and also in https://doi.org/10.5500/wjt.v11.i11.480. In this last article every case of Collapsing Glomerulopathy after Sars-Cov2 infection was recorded in a table. Both references have been added to the article. On the other hand, thanks to your suggestion, we have added two tables summarizing the five cases of Collapsing Glomerulopathy after Sars-Cov2 vaccination previously published.
- The patients did not have any other comorbidities. He was not tested for Sars-Cov2 infection on the onset of the nephrotic syndrome, but serologies for Sars-Cov2 both anti-N and anti-S IgG were negative on the follow-up, and IgG producing mBc were not detected. As referenced, mostly all patients elicit humoral response shortly after Sars-Cov2 infection, so these findings suggest that this case was not secondary to an infection.
- The determination of the APOL1 risk variants has been clarified. Our patient was G0/G0, so no risk variants were identified. As pointed out in https://doi.org/10.1681/asn.2021070931 (added as a reference now) most but not all patients with collapsing glomerulopathy following Sars-Cov2 infection do have risk alleles. Besides, one of the patients in the literature developing CG after Sars-Cov2 vaccination was G1/G0, a genotype not associated with renal diseases. Therefore, COVAN is a term that refers to every Collapsing Glomerulopathy after Sars-Cov2 vaccination, independently of the presence of APOL1 risk variants to develop it (although most of the patients do present them).
Thank you very much
Round 2
Reviewer 2 Report
The suggested changes have been made.